

# Differences in small noncoding RNAs profile between bull X and Y sperm

Hao Zhou[1,2,*], Jiajia Liu[3,*], Wei Sun[2], Rui Ding[2], Xihe Li[2], Aishao Shangguan[1], Yang Zhou[1], Tesfaye Worku[1], Xingjie Hao[1], Faheem Ahmed Khan[4], Liguo Yang[1] and Shujun Zhang[1]

[1] Key Laboratory of Agricultural Animal Genetics, Breeding and Reproduction, Education Ministry of China, College of Animal Science and Technology, Huazhong Agricultural University, Wuhan, China
[2] Inner Mongolia Saikexing Institute of Breeding and Reproductive Biotechnology in Domestic Animal, Hohhot, China
[3] School of Biological Science and Technology, University of Jinan, Jinan, China
[4] Department of Zoology, University of Central Punjab, Lahore, Pakistan
[*] These authors contributed equally to this work.

Corresponding author
Shujun Zhang, marywood@163.com, sjxiaozhang@mail.hzau.edu.cn

## ABSTRACT

The differences in small noncoding RNAs (sncRNAs), including miRNAs, piRNAs, and tRNA-derived fragments (tsRNAs), between X and Y sperm of mammals remain unclear. Here, we employed high-throughput sequencing to systematically compare the sncRNA profiles of X and Y sperm from bulls ($n = 3$), which may have a wider implication for the whole mammalian class. For the comparison of miRNA profiles, we found that the abundance of bta-miR-652 and bta-miR-378 were significantly higher in X sperm, while nine miRNAs, including bta-miR-204 and bta-miR-3432a, had greater abundance in Y sperm ($p < 0.05$). qPCR was then used to further validate their abundances. Subsequent functional analysis revealed that their targeted genes in sperm were significantly involved in nucleosome binding and nucleosomal DNA binding. In contrast, their targeted genes in mature oocyte were significantly enriched in 11 catabolic processes, indicating that these differentially abundant miRNAs may trigger a series of catabolic processes for the catabolization of different X and Y sperm components during fertilization. Furthermore, we found that X and Y sperm showed differences in piRNA clusters distributed in the genome as well as piRNA and tsRNA abundance, two tsRNAs (tRNA-Ser-AGA and tRNA-Ser-TGA) had lower abundance in X sperm than Y sperm ($p < 0.05$). Overall, our work describes the different sncRNA profiles of X and Y sperm in cattle and enhances our understanding of their potential roles in the regulation of sex differences in sperm and early embryonic development.

# INTRODUCTION

Several previous studies have considered the question of diversity between X and Y sperm, demonstrating significant variation in their structure, morphology, motility, and energy metabolism (*Cui & Matthews, 1993*; *Sarkar et al., 1984*; *Shettles, 1960*). With the advent of computer-assisted sperm analysis, which allows the objective evaluation of kinetic parameters, most of the detected variations have been considered controversial

(*Hossain, Barik & Kulkarni, 2001*; *Penfold et al., 1998*). Sperm carry different sex chromosomes (either an X- or a Y-chromosome), which provide clues for the discovery of other differences between X and Y sperm. Indeed, such differences have been identified in several different types of profiles (*Yadav et al., 2017*), including protein profiles (*Chen et al., 2012*; *De Canio et al., 2014*) and messenger RNA (mRNA) profiles (*Chen et al., 2014*), by using high-throughput measurement technologies. However, whether there are differences in the profiles of sncRNAs is still unknown.

Recently, an increasing number of studies in several species have shown that mature ejaculate sperm carry thousands of sncRNA, including tRNA-derived fragments (tsRNAs), ribosomal RNAs (rRNAs) and small nucleolar RNAs (snoRNAs), especially microRNAs (miRNA) and Piwi-interacting RNAs (piRNA) (*Sellem et al., 2020*). As the best-studied type of small noncoding RNA, miRNA has been implicated in posttranscriptional control, by binding to an Argonaute (AGO) protein to stabilize its target through binding to the 3′ untranslated region(UTR), and in the regulation of translation by targeting amino acid coding (CDS) regions (*Hausser et al., 2013*). Even though the complexity of sperm miRNA has been well characterized in several mammalian species, including humans (*Pantano et al., 2015*; *Tay et al., 2008*), mice (*Nixon et al., 2015*), pigs (*Chen et al., 2017*; *Zhang et al., 2017*) and bulls (*Capra et al., 2017*), the functions of most of the sperm miRNAs remain enigmatic. On one hand, sperm miRNAs seem to be required by the sperm themselves and may have a function that impacts sperm motility. As a snapshot of what remains after spermatogenesis, the sperm miRNA profile was shown to be altered in different types of motile sperm in bulls (*Capra et al., 2017*). On the other hand, sperm miRNAs may play roles in processes, such as fertilization (*Yuan et al., 2016*) and, subsequently embryonic development (*Yuan et al., 2016*), potentially even transmitting paternally acquired phenotypes (*Grandjean et al., 2015*; *Rodgers et al., 2015*; *Sharma et al., 2015*) after they are carried into the fertilized oocyte. The major reason why sperm miRNAs execute these roles is that in the fertilized oocyte they regulate target maternal mRNAs (*Rodgers et al., 2015*; *Wang et al., 2017*). miRNAs carried by sperm may control maternal mRNAs expression levels to affect epigenetic reprogramming, the cleavage, and apoptosis of somatic cell nuclear transfer (SCNT) embryos in cattle (*Wang et al., 2017*). Transgenerational effects associated with parental diet were also proposed to be mediated, at least in part, by sperm tsRNAs (*Chen et al., 2016*). In contrast to miRNAs (∼22 nt), piRNAs (24–31 nucleotides) are marginally longer, expressed primarily in the germline and binding to the Piwi class as compared to the Ago-class (*Aravin et al., 2006*; *Girard et al., 2006*; *Siomi et al., 2011*). piRNAs are assumed to be produced from the polycistronic RNAs that are transcribed in the genome from a small number of specific regions named piRNA clusters. It has been proposed that piRNAs may protect genome integrity from the deleterious effects of repetitive and transposable elements by binding to the elements (*Krawetz et al., 2011*). piRNAs were shown to be the more abundant set of regulatory sncRNAs than other types of sncRNAs in human sperm (*Pantano et al., 2015*), their expression in sperm correlated to sperm concentration and fertilization rate (*Cui et al., 2018*). These groundbreaking detection of sperm sncRNA led us to question whether there are any differences in the

sncRNA profiles of two types of sperm, and if so, what are the specific functions of sncRNA showing differences between X and Y sperm?

Here, we systematically compared the abundance of several kinds of sncRNA species between X and Y sperm, especially in miRNAs, piRNAs, and tsRNAs. To explore the roles that differentially abundant (DA) miRNAs play in sperm and fertilized oocyte, we predicted their target binding sites in 3′ UTRs and CDS regions and performed Gene Ontology (GO) and Kyoto Encyclopedia of Genes and Genomes (KEGG) pathway enrichment analyses of the target genes presented in sperm and mature oocyte. To our knowledge, this study provides the first description of the differences in the sncRNA profiles of X and Y sperm, which could improve our understanding of their possible functions in the regulation of sex differences in sperm and early embryonic development.

## MATERIALS & METHODS

### Bull X and Y sperm collection

Semen samples were obtained from the Saikexing Institute (Hohhot, China). Briefly, samples were collected from three Holstein bulls at three years of age. These bulls were fed the same diet daily and reared in the same conditions and environments. The semen was sampled from them using an artificial vagina and stored at room temperature (18 °C) for 1 h. Subsequently, the samples were passed through a 50 μm filter to remove debris or clumped sperm, and the sperm were stained with the Hoechst-33342 fluorophore (Sigma, St Louis, USA) via incubation at 34 °C for 45 min in darkness. After staining, the three sperm samples were separated into three X sex-sorted semen and three Y sex-sorted semen using a high-speed MoFlo SX XDP flow cytometer (DakoCytomation, Fort Collins, USA). The purity of X and Y sex-sorted semen were tested by using the sort reanalysis method (*Welch & Johnson, 1999*). In brief, 20 μl semen from each sample sonicated to remove the sperm tail and stained with 20 μl Hoechst-33342 fluorophore via incubation at 34 °C for 20 min in darkness. These samples were input into the high-speed MoFlo SX XDP flow cytometer to measure the purity of X and Y sex-sorted semen by performing the resorting procedure. Then, the sorted semen was washed twice in phosphate buffered saline (PBS, GE Healthcare Life Sciences, USA) through centrifugation at 700 g for 10 min at 20 °C. We removed the supernatants and mixed the sperm pellets with TRIzol by incubation for 5 min at RT (Sigma; 0.5 ml per $1 \times 10^7$ sperm). After that, the samples were store on dry ice for next day use. The Scientific Ethics Committee of Huazhong Agricultural University approved the experimental design and animal treatments for the present study (permit number: HZAUSW-2017-012), and all experimental protocols were conducted in accordance with the guidelines.

### Sperm total RNA isolation, libraries preparation, and sequencing

The TRIzol method has been used to extract the sperm total RNA of the six samples (*Das et al., 2010*). Comprehensive protocols were outlined in our earlier report (*Shangguan et al., 2020*). Using a 2100 Bioanalyzer, the RNA integrity was assessed after extraction of the RNA (Agilent Technologies, USA). The validated RNAs of X and Y sperm (10 ng RNA from each sample) were sequenced on the BGISEQ-500 platform (*Fehlmann et al., 2016*)

at the BGI company (Shenzhen, China). A sequencing library was prepared and sequenced according to a standard protocol established by the BGI (*Fehlmann et al., 2016*).

## Preprocessing of small noncoding RNA data

After sequencing, the raw data were obtained from X and Y sperm samples. The quality of sequencing reads was tested by fastQC (http://www.bioinformatics.babraham.ac.uk/projects/fastqc/). The adapters were initially detached from the raw sequence data (3′ adapters: AGTCGGAGGCCAAGCGGTCTTAGGAAGACAA, 5′ adapters: GAACGACATGGCTACGATCCGACTT). Also, in order to obtain clean data, we used Trimmomatic to trim the low-quality bases of each sequence (*Bolger, Lohse & Usadel, 2014*). The options below were used to trim: SLIDINGWINDOW 4:15, MINLEN 15, MAXINFO 15:0.8. The sequencing data of X and Y sperm sncRNA have been deposited in the Sequence Read Archive (https://www.ncbi.nlm.nih.gov/sra) with the accession number PRJNA624261.

## Small noncoding RNA annotation

Analysis of the small noncoding RNA data was performed using Unitas with options: -tail 1 -mismatch 0 (*Gebert, Hewel & Rosenkranz, 2017*) Unitas is a software for the classification and annotation of mature miRNAs, rRNAs, piRNAs, tsRNAs, protein-coding RNAs, small nucleolar RNAs (snoRNAs), small nuclear RNAs (snRNAs), low complexity RNAs, non-annotated RNAs, and miscellaneous RNAs (miscRNAs). To reduce false-positive results for miRNA annotation, Mirdeep2 (*Friedländer et al., 2011*) was also used to predict putative known mature miRNAs of *Bos taurus*. We chose 17-23 nt sequences (based on the most of the length of mature miRNA (*Sendler et al., 2013*)) from the clean data from which the *Bos taurus* rRNA and tRNA sequences had been removed as the input for Mirdeep2. The known mature *Bos taurus* miRNAs dataset (miRBase v.21, http://www.miRbase.org/) was used for miRNA detection. Then, the relative abundance of all miRNAs annotated from two software was standardized to the transcripts per million reads value (RPM) according to the formula: RPM = (mapped reads $\times 10^6$)/total reads, and miRNAs with an RPM >5 that were found in at least 2 samples were identified as miRNAs expressed.

The information of piRNA cluster and tsRNAs of X and Y sperm was obtained from the output files of Unitas (version 2.1) (*Rosenkranz & Zischler, 2012*). We obtained piRNA clusters data of each sample. For the piRNA cluster, if the genomic regions of two clusters identified in one sample were overlapped on the genome, they were considered as the same cluster. The same clusters found in all 3 replications were identified as the conserved cluster in each group. Genes and repeats falling within the detected clusters were retrieved using bovinemine (http://bovinegenome.org). Furthermore, reads mapped within the detected clusters were also retrieved to map to all available piRNA database on piRBase (http://regulatoryrna.org/database/piRNA/download.html) to identify the putative piRNA using the Bowtie software (*Langmead et al., 2009*). piRNAs with an RPM >10 that were annotated in at least three samples were defined as expressed piRNAs.
## Analysis of differential miRNA abundance

Differentially abundant analysis was carried out using the Bioconductor DEseq2 R package (*Love, Huber & Anders, 2014*). By applying thresholds of a $P$-value <0.05 and |log2(fold change)|>1, the remaining miRNAs were defined as significantly differentially abundant (DA) miRNAs. Furthermore, the analysis of the differential abundance of piRNAs and tsRNAs between X and Y sperm was the same as that of miRNAs.

## Functional annotation of DA miRNAs

Two datasets named transcriptome data sequenced from single bull metaphase II oocyte (GSE59186) ($n = 2$) and Bull sperm transcriptome data(SRA055325) ($n = 1$) that have been earlier published were applied to explore the function of X and Y sperm DA miRNAs. The raw sequencing data were collected from the Sequence Read Archive (SRA) and were reanalyzed following these processes: (1) We used Cutadapt (https://cutadapt.readthedocs.io/en/stable/) to cut the sequencing adapters and FASTX-Toolkit (http://hannonlab.cshl.edu/fastx_toolkit/) was used to filter sequences of low quality with the options "fastq_quality_trimme $r - v - Q$ 33 -l $30 - i - t$ 20"; (2) Clean reads were aligned to the reference genome (Btau_4.6.1) by Tophat software (*Trapnell et al., 2012*); (3) For each gene model the mapped sequencing was counted and recorded via Cufflinks in Fragments Per Kilobase Million (FPKM) (*Trapnell et al., 2012*). Moreover, sperm transcripts were filtered with FPKM <50, and oocyte genes with FPKM >50 were retained in at least one sample. Ultimately, we obtained 1,036 sperm genes and 2,584 oocyte genes (Table S5). miRwalk was applied to identify the targets of the DA miRNAs with TarPmiR-algorithms (http://129.206.7.150/) (*Dweep & Gretz, 2015*). miRNA binding sites including CDS and 3′ UTR within the complete sequences of all *Bos taurus* genes were investigated. Only target genes with binding $P$-values >0.8 were retained for further analysis. The mature oocyte and sperm gene sets were further overlapped with the target genes set of DA miRNAs.

Functional annotations of the target genes found in sperm and matured oocyte were carried out by Clusterprofiler software, respectively (*Yu et al., 2012*). Genes acquired were subjected to enrichment analyses by GO and KEGG to detect the significantly enriched terms in target genes. Also, terms with an adjusted $p < 0.05$ by Benjamini–Hochberg (BH) multiple testing were deemed significant.

## Quantitative real-time PCR (qPCR) validation of the sequencing results

To verify the accuracy of high throughput sequencing results, we randomly selected and confirmed the abundance of four miRNAs (bta-miR-204, bta-miR-3432a, bta-miR-652, and bta-miR-378) in X and Y sperms by qPCR. Sperm RNA was produced from another three bulls semen following the aforementioned protocol. Using the miScript II RT Kit (Qiagen), total RNA of each sample was reverse-transcribed into cDNA. qPCR was carried out on an ABI 7500 Real-Time PCR system (Applied Biosystem) by using miScript SYBR Green PCR Kit (Qiagen) with a miRNA-specific forward primer. The relative abundant levels of the miRNAs were normalized to U6 and calculated by $2^{(-\Delta\Delta Ct)}$ approach. Table S11 shows the primer information.

## RESULTS

### Evaluation of sperm RNA quality

After sorting, we obtained around 0.9 billion sex-sorted sperm per sample for RNA extraction. Information for the sex-sorted semen samples and sperm RNA quality data are shown in Table S1. It reveals two well-known characteristics of sperm RNA (an absence of intact ribosomal RNA (rRNA) and predominance of short-length RNA molecules) (*Johnson et al., 2011*). All the samples exhibited 28S/18S values of 0, indicating a lack of intact foreign RNA in sperm RNA samples. The RNA integrity number (RIN) was approximately 2.5, which was conformed to the characteristic of RNA in sperm (*Mao et al., 2014*; *Sendler et al., 2013*; *Yuan et al., 2016*).

### Read counts of each RNA class

After sequencing, we obtained 33,974,607, 30,027,769 and 30,422,125 raw reads (31,474,834 $\pm$ 2,173,828, mean $\pm$ SD) for X4069, X4118 and X4131 sample, respectively and 30,532,748, 29,907,341, and 29,594,201 raw reads (30,011,430 $\pm$ 477,853) for Y4069, Y4118 and Y4131 sample, respectively. The quality control results show that bases of these raw sequences are with high quality score and the raw sequence lengths are 50 nt in all samples, suggesting the good quality of sequencing data we obtained (Additional material 1). After removing low-quality reads, 25,037,734, 26,837,611, and 24,848,674 clean reads (25,574,673 $\pm$ 1,097,814) for X4069, X4118 and X4131 sample, respectively, and 28,164,513, 27,819,402, and 26,946,322 clean reads (27,643,412 $\pm$ 627,875) for Y4069, Y4118 and Y4131 sample, respectively, remained. The proportion of *Bos taurus* miRNA (11.9% vs. 20.2%, $t$-test, $p = 0.003786$) and non-annotated sequence (68.1% vs. 52.7%, $t$-test, $p < 0.0001$) were significantly different between X and Y sperm, while the other small noncoding RNA species (miRNAs of other species, rRNAs, tsRNAs, protein-coding RNAs, snoRNAs, miscRNAs, snRNAs, and piRNAs) presented similar proportions (Table S1).

### Identification of DA tsRNAs in X and Y sperm

We identified 52 tsRNAs in X and Y sperm (Table S10), The comparison of X and Y sperm revealed only 2 significantly DA tsRNAs, including tRNA-Ser-AGA and tRNA-Ser-TGA, both they have lower abundance in the X sperm ($p < 0.05$).

### Identification of DA miRNAs in X and Y sperm

In total, 490 known *Bos taurus* miRNAs were detected by Unitas, and 202 known *Bos taurus* miRNAs were identified by Mirdeep2, respectively. Among these miRNAs, 49 and 21 miRNAs were differentially abundant at a significant level ($p < 0.05$ and |log2fold change|>1) (Table S2). We obtained 12 DA miRNAs that overlapped between 49 and 21 DA miRNAs identified by two different software, including 3 highly abundant miRNAs in X sperm (bta-miR-15a, bta-miR-652, and bta-miR-378) and 9 more enriched miRNAs in Y sperm (bta-miR-204, bta-miR-1271, bta-miR-211, bta-miR-375, bta-miR-3432a, bta-miR-127, bta-miR-6529a, bta-miR-369-5p, and bta-miR-196a) (Table 1). Among these DA miRNAs, bta-miR-204, bta-miR-375, and bta-miR-378 were the most significantly DA miRNAs ($p < 0.005$). Bta-miR-204 (log2FC $= -2.36$, $P = 0.0002$) was the most

**Table 1  Summary of differentially abundant miRNAs between X and Y sperm.**

| miRNA | Log$_2$FC | P-value | X (RPM) | Y (RPM) | Location (Chr: Start..End) |
|---|---|---|---|---|---|
| bta-miR-204 | −2.36 | 0.0001 | 44289.4 | 159757.7 | 8:47259591..47259612 |
| bta-miR-1271 | −2.03 | 0.0202 | 30.3 | 72.4 | 7:39194962..39194983 |
| bta-miR-211 | −1.98 | 0.0076 | 66.9 | 153.3 | 21:28046473..28046493 |
| bta-miR-375 | −1.76 | 0.0021 | 146.2 | 328.7 | 2:107667524..107667546 |
| bta-miR-3432a | −1.63 | 0.0164 | 2037.4 | 3111.0 | 21:55746836..55746857 |
| bta-miR-127 | −1.58 | 0.0262 | 49.0 | 78.3 | 21:67429798..67429819 |
| bta-miR-6529a | −1.43 | 0.0240 | 1411.7 | 2695.0 | 1:65453397..65453417 |
| bta-miR-369-5p | −1.34 | 0.0302 | 116.5 | 188.6 | 21:67603550..67603569 |
| bta-miR-196a | −1.33 | 0.0292 | 129.9 | 205.0 | 19:38497007..38497028 |
| bta-miR-15a | 1.47 | 0.0173 | 161.5 | 32.6 | 12:19596395..19596415 |
| bta-miR-652 | 2.33 | 0.0191 | 293.4 | 27.9 | 30:62939343..62939363 |
| bta-miR-378 | 4.11 | 0.0000 | 50.7 | 1.6 | 4:10715305..10715326 |

**Notes.**

Log$_2$FC refers to Log$_2$ Fold Change, *X* and *Y* value refer to the average RPM of miRNA abundance for 3 replications of X and Y sperm. The Log$_2$FC, *P*-value, and RPM values are calculated from the results annotated by Unitas (Table S2).

abundant miRNA in both fractions and was previously identified in human, pig, and mouse epididymal sperm (Table 2). The abundance of bta-miR-652 (log2FC = 2.26, $P = 0.0092$) that was greater in X sperm was the only DA miRNA detected on the X chromosome (Table 1), whose abundance has been detected in human and mouse epididymal sperm but not in boar sperm (Table 2) (*Nixon et al., 2015*; *Pantano et al., 2015*). Furthermore, the DA miRNAs (4/12) showed a higher preference for chromosome 21 (Table 1).

In addition to the 12 DA miRNAs, another 118 *Bos taurus* miRNAs included the 10 most abundant non -DA miRNAs between X and Y sperm: bta-miR-100, bta-let-7a-5p, bta-miR-22-3p, bta-miR-151-5p, bta-miR-21-5p, bta-miR-99a-5p, bta-miR-16b, bta-miR-7, and bta-miR-27a-3p (Table S2). Interestingly, these miRNAs together accounted for 92% of the RPM values of all non-DA miRNAs, indicating that the levels of the miRNAs differed sharply. We also compared our findings with recent studies and found that bta-miR-100 is also present at high levels in porcine, bull, human, and mouse epididymal sperm miRNA profiles (*Capra et al., 2017*; *Chen et al., 2017*; *Nixon et al., 2015*; *Pantano et al., 2015*) (Table 2), suggesting that its biological functions are conserved across these species.

### Prediction of DA miRNA target genes in mature oocyte and sperm

We predicted target sites in the 3′ UTRs and CDS regions, of 1,677 and 4,028 genes, respectively, for nine upregulated DA miRNAs in Y sperm, and in 510 and 1,224 genes, respectively, for three upregulated DA miRNAs in X sperm (Fig. 1, Table S4). We found that a greater number of target genes were predicted to be bound in CDS regions than in 3′ UTRs.

Similar to the prediction results, the number of predicted target genes present in mature oocyte and sperm that were bound at CDS sites was greater than the number bound at 3′ UTR sites. In the mature oocyte gene sets (Table S5), 602 and 248 genes were targeted by nine upregulated DA miRNAs at CDS and 3′ UTR binding sites, respectively, in Y sperm.
**Table 2  Comparison of differentially abundant (DA) and highly conserved (Non-DA) miRNAs identified in our study with other studies.**

| miRNA (*Bos Taurus*) | Reads per million | | DE | Non-DE | Previously identified | | | | | |
|---|---|---|---|---|---|---|---|---|---|---|
| | X | Y | | | Bull[1] (Top 20) | Bull[2] (Top 20) | Bull[2] (All) | Human | Boar | Mouse |
| miR-100 | 128,833 | 99,447 | − | + | + | + | + | + | + | + |
| let-7a-5p | 15,372 | 13,679 | − | + | + | − | − | + | − | + |
| miR-22-3p | 18,459 | 9,877 | − | + | − | + | + | + | + | + |
| miR-151-5p | 16,214 | 11,325 | − | + | − | + | + | − | + | + |
| miR-21-5p | 15,621 | 5,607 | − | + | + | + | + | + | − | + |
| miR-449a | 8,210 | 9,565 | − | + | − | − | − | + | − | + |
| miR-99a-5p | 9,363 | 6,399 | − | + | + | − | − | + | − | + |
| miR-16b | 9,903 | 3,619 | − | + | − | + | + | − | − | − |
| miR-7 | 4,595 | 6,283 | − | + | + | − | − | − | + | − |
| miR-27a-3p | 7,530 | 2,925 | − | + | − | + | + | + | − | + |
| miR-127 | 49 | 78 | + | − | − | − | − | + | + | + |
| miR-1271 | 30 | 72 | + | − | − | − | − | − | − | − |
| miR-15a | 162 | 33 | + | − | − | − | − | + | + | + |
| miR-196a | 130 | 205 | + | − | − | − | − | + | + | + |
| miR-204 | 44,384 | 159,903 | + | − | + | + | + | + | + | + |
| miR-211 | 67 | 153 | + | − | + | − | − | − | − | − |
| miR-3432a | 2,042 | 3,114 | + | − | − | − | − | − | − | − |
| miR-369-5p | 117 | 189 | + | − | − | − | − | − | − | − |
| miR-375 | 147 | 329 | + | − | + | + | + | + | − | + |
| miR-378 | 51 | 2 | + | − | − | − | − | − | − | + |
| miR-652 | 295 | 28 | + | − | − | − | − | − | − | + |
| miR-6529a | 1,415 | 2,697 | + | − | − | − | − | − | − | − |

**Notes.**
'+' and '−' refer to the miRNAs presented and absented in the datasets of DE miRNAs, non-DE miRNAs, Bull[1] (*Sellem et al., 2020*), Bull[2] (*Capra et al., 2017*), Bull[2] (top 20) (*Capra et al., 2017*), Human (*Pantano et al., 2015*), Boar (*Chen et al., 2017*), and Mouse (*Nixon et al., 2015*).

Three upregulated DA miRNAs in X sperm targeted 163 and 70 genes in CDS and 3′ UTR binding sites, respectively. In contrast, in the sperm gene sets (Table S4), nine upregulated DA miRNAs of Y sperm targeted 146 and 37 genes at CDS and 3′ UTR binding sites, respectively, and three upregulated DA miRNAs of X sperm targeted 33 and 11 genes at CDS and 3′ UTR binding sites, respectively (Fig. 1, Table S6). Taken together, the results suggested that the phenomenon of miRNA regulation of gene expression through CDS regions may be widely present in sperm and fertilized oocyte. In addition, the DA miRNAs may exhibit one or more target genes, due to interacting with different target regions (*Hausser et al., 2013*). By removing these repeated target genes, we eventually obtained 887 target genes in the mature oocyte and 210 target genes in sperm (Fig. 1). Among these target genes, 79% (697/887) and 82% (172/210) were bound at CDS regions in mature oocyte and sperm, respectively (Table S6). Furthermore, we found that 6.3% and 23.3% of oocyte genes were targeted by X and Y sperm upregulated DA miRNAs, respectively, on CDS sites, which was greater than 3.2% and 14.1% of sperm genes targeted by DA miRNAs on CDS sites (Fig. 1). Similarly, on the 3′ UTR sites, 2.7% and 9.6% of oocyte genes were targeted by X and Y sperm highly abundant miRNAs, respectively, which was higher than
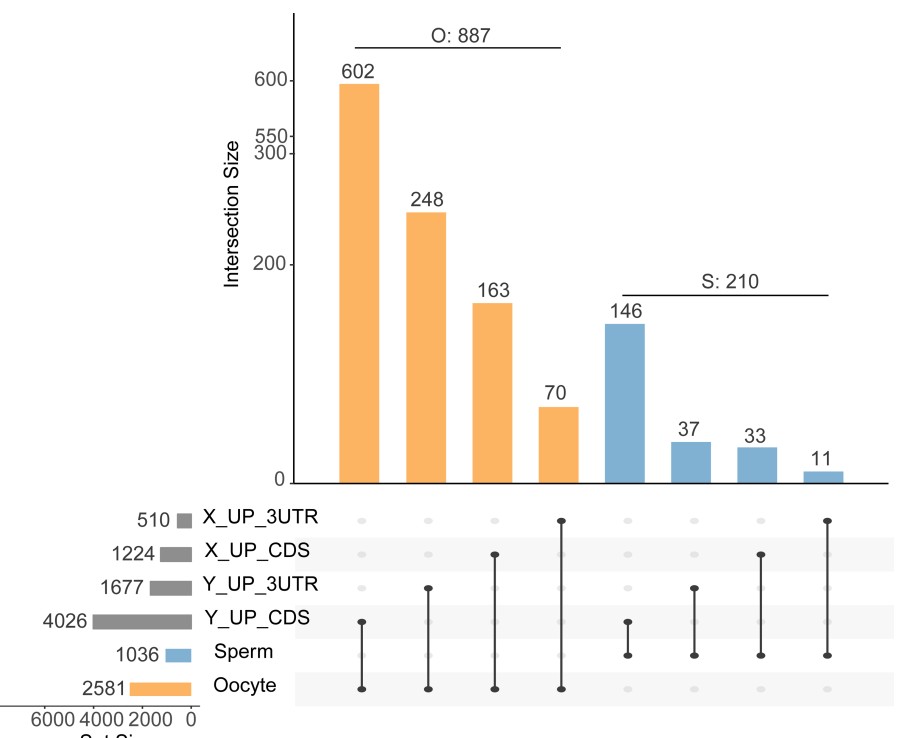

**Figure 1** **Upset plot of intersections between gene set targeted by differentially abundant miRNA through the 3′ UTR region and CDS region and sperm or matured oocyte gene set.** The horizontal bars indicate target (dark gray) sperm (blue), and matured oocyte (yellow) gene sets. The vertical bar chart indicates the intersection size. The blue vertical bars refer to the overlapping target genes in sperm, the yellow vertical bars refer to target genes in the matured oocyte. The number over the bar indicates the number of genes. Dark connected dots indicate which substrates are considered for each intersection. X/Y_UP_3UTR/CDS refer to genes targeted by up-regulated DA miRNAs in X/Y sperm through binding the 3UTR/CDS region

1.1% and 3.6% of sperm genes targeted by DA miRNAs (Fig. 1). These results suggested that DA miRNAs were more prone to target genes in matured oocyte than in sperm, which was consistent with the previous finding that miRNA targets are likely absent in sperm (*Krawetz et al., 2011*).

## Functional analysis of DA miRNA targets in sperm and mature oocyte

In the functional enrichment analysis of the 210 targets in sperm, the top significantly enriched GO categories were mainly related to mRNA processing, nucleosome binding, and nucleosomal DNA binding (adjusted $p < 0.05$, Fig. 2A, Table S7). Surprisingly, the 887 targets in mature oocyte were significantly related to 11 catabolic processes, including macromolecule catabolic processes, cellular protein catabolic processes, and organonitrogen compound catabolic processes (adjusted $p < 0.05$, Fig. 2A, Table S7). The 17 genes (*MAGOH, PSMC5, DICER1, KCTD13, TRIP12, EIF3E, PSMA5, UBE2H, PSMD11, USP1, SKP1, ARIH1, EZR, IDE, TIMP3*, and *TRIM13*) related to catabolic processes were

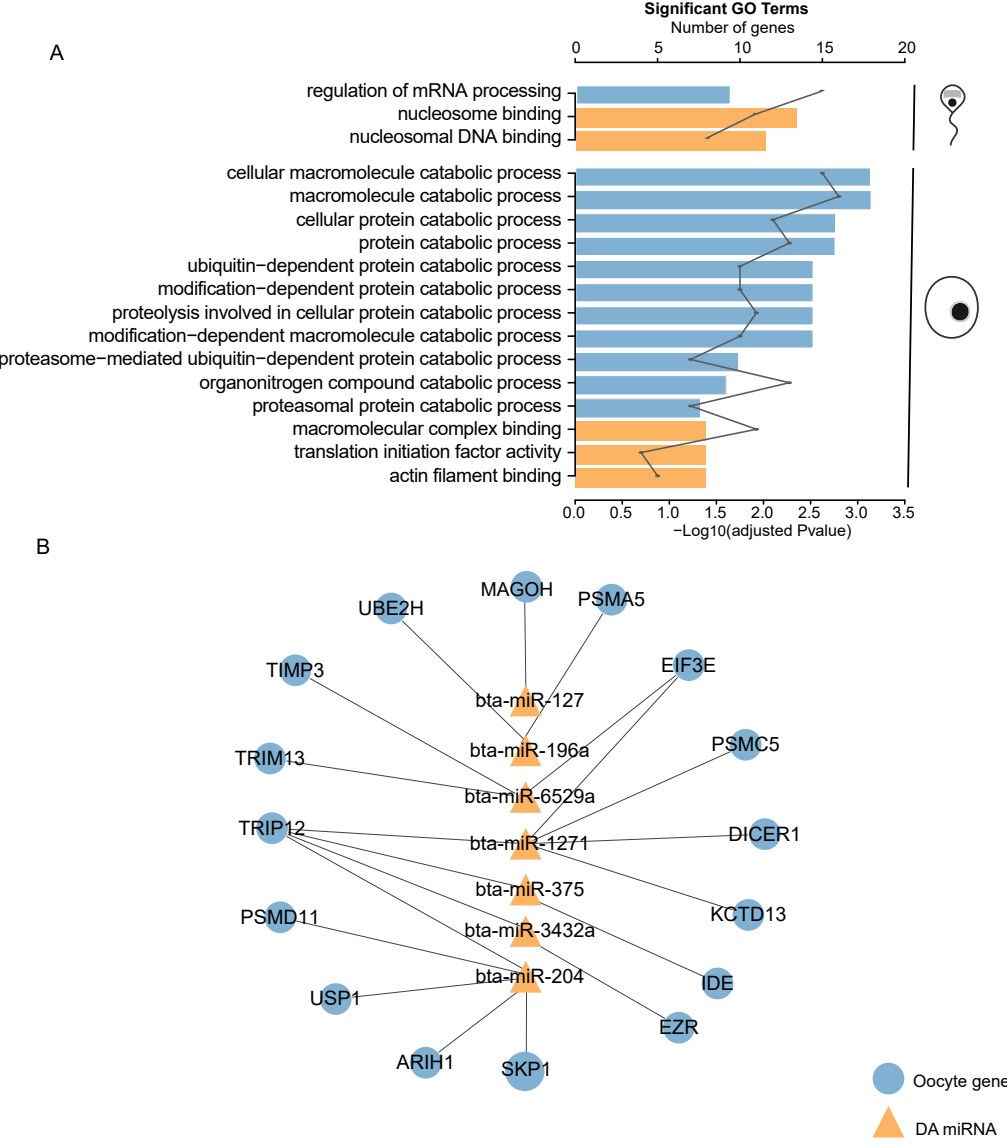

**Figure 2** **Functional analysis of differentially abundant miRNAs with their predicted target genes presented in sperm and mature oocyte.** (A) Bar plot shows the significant (adjusted $P$-value $< 0.05$) gene ontology (GO) terms enriched by the target genes in sperm and mature oocyte. Bar height shows the enrichment scores ($-$log adjusted $P$-value) of the GO terms. Line plot depicts the number of genes that belong to each category. Blue and yellow bar show GO terms of biological process and cellular component, respectively. (B) Interactions between DA miRNAs and their mature oocyte target genes involved in catabolic processes.

targeted by seven Y sperm upregulated DA miRNAs: bta-miR-127, bta-miR-1271, bta-miR-196a, bta-miR-204, bta-miR-3432a, bta-miR-375 and bta-miR-6529a (Fig. 2B, Table S7). This finding may indicate that these DA miRNAs and their targets in the oocyte are involved in a series of catabolic processes. Here, no enriched GO terms overlapped between the sperm and mature oocyte (Fig. 2A).

On the other hand, in the KEGG analysis of the 210 targets in sperm, seven KEGG pathways were significantly enriched, including the cell cycle and RNA transport (adjusted $p < 0.05$, Table S7). In contrast, only the endometrial cancer pathway was significantly enriched by 887 targets in the mature oocyte (adjusted $p < 0.05$, Table S7).

**Different distributions of piRNA cluster between X and Y sperm**

Here, a total of 21 and 12 unique piRNA clusters loci were identified in X and Y sperm, respectively, and 71 clusters loci were shared between two fractions, cluster XY69 that was located in the region of 25,287,459bp to 25,335,935bp on chromosome 28 was reported to be conserved deeply in Eutherian mammals (Table S8).

To best understand the potential functions of piRNAs in X and Y sperm, we searched genes and transposons falling within their unique cluster loci. 362 repeats and 14 genes were within the unique cluster region of X sperm. Of them, BovB, Bov-tA3, and Bov-A2 were the top three repeat elements detected in unique piRNA clusters of X sperm. Cluster X9 located on 14 chromosomes contained the greatest number of repeat elements (102) in all the clusters identified (Fig. 3A, Table S9). 14 genes enriched in 14 GO terms (adjusted $p < 0.05$), including biology function of galactosyl ceramide catabolic process, and galactolipid metabolic process. On the other hand, 169 repeat elements and 7 genes are within unique cluster loci of X sperm, among them, 23 BovB was identified, which was the greatest number of repeat elements identified, followed by Bov-tA2 (21) and BOV-A2 (13). Cluster Y11 located on chromosome 24 contained the greatest number of 77 repeat elements (Fig. 3A, Table S9). 13 GO terms were enriched by 7 genes (adjusted $p < 0.05$), including nucleosome assembly and histone H3-K27 trimethylation. Details of X and Y unique piRNA clusters including genes, repeats, and function of genes falling within the cluster regions are given in Fig. 3A and Table S9. Furthermore, the expressed piRNAs in X and Y sperm were explored, we identified 582 piRNAs. Of them, 28 piRNAs were differentially abundant ($p < 0.05$), 15 piRNAs had higher abundance in Y sperm and 13 piRNAs were enriched in X sperm. The most significantly enriched piRNAs in X and Y sperm are piR-5346348 ($p = 1.22 \times 10^{-12}$) and piR-5342466 ($p = 1.13 \times 10^{-12}$), respectively (Fig. 3B, Table S10).

**qPCR validation of DA miRNAs**

To validate the high-throughput sequencing results, we randomly selected four miRNAs (bta-miR-204, bta-miR-3432a, bta-miR-652, and bta-miR-378) to perform the qPCR experiment. The relative fold changes of these miRNAs in qPCR were concordant with the sequencing results (Fig. 4, Table S11), indicating that the miRNA identification and abundance estimation were reliable.

## DISCUSSION

In mammals, X sperm contains more DNA than Y sperm, and these DNA differences might result in differences in small RNA abundance. In our study, the differential abundance of miRNAs, piRNAs, and tsRNAs between the two types of sperm were identified. Previous studies have revealed that adjacent sperm cells can share gene products through intercellular

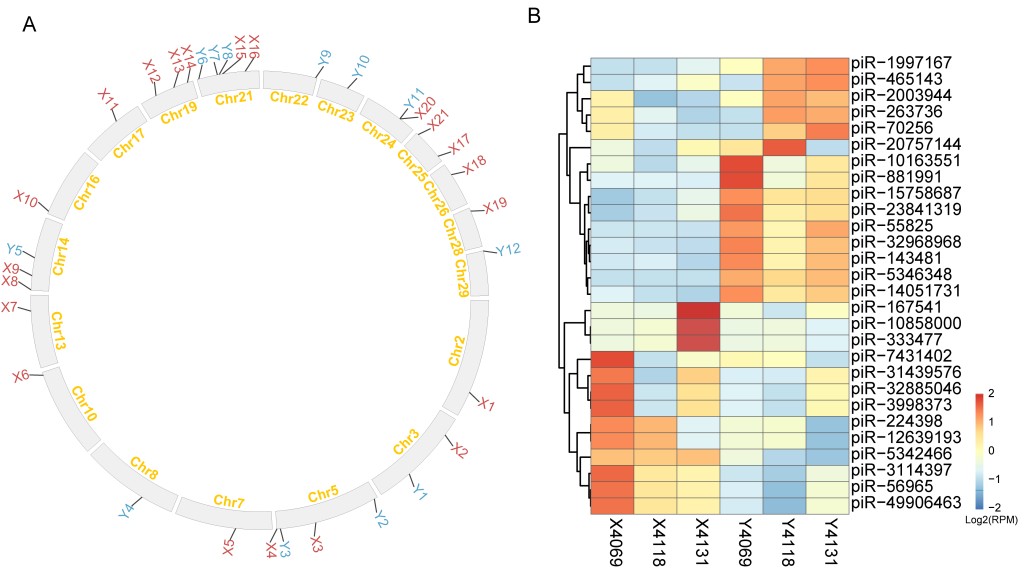

**Figure 3** **piRNA clusters and piRNAs of X and Y sperm.** (A) The distribution of piRNA clusters along the chromosomes. The label of X and Y sperm unique clusters are shown by red and blue, respectively. (B) Heatmap of clustering to 28 DA piRNAs detected.

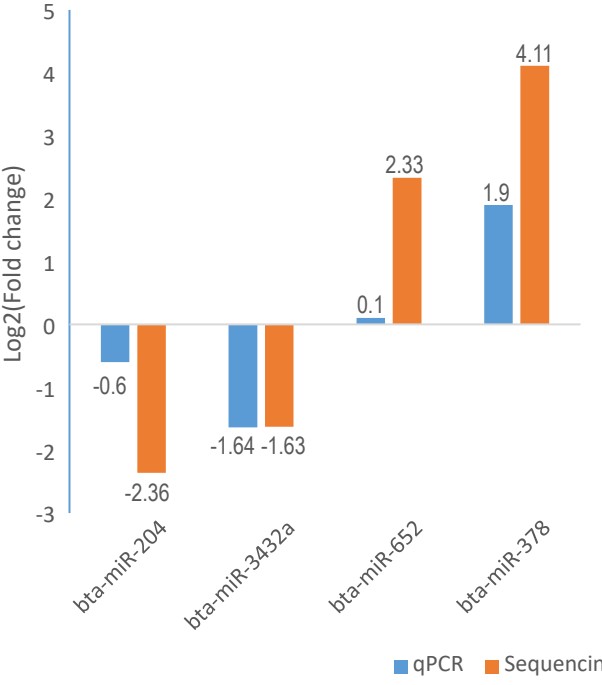

**Figure 4** **Illustration of the qPCR results for four selected differentially abundant miRNAs.** The *X*-axis represents four selected miRNAs and *Y*-axis represents the log2 fold change (X sperm/Y sperm) obtained from qPCR and sequencing.

bridges during spermatogenesis, suggesting that sncRNA molecules may be shared between X and Y sperm cell during spermatogenesis (*Fawcett, Ito & Slautterback, 1959*) and retained within mature sperm, which may explain why a part of non-DA small RNAs was identified between X and Y sperm. However, these products are probably not all shared through the intercellular bridge (*Ventelä, Toppari & Parvinen, 2003*). Indeed, differences between X and Y sperm have been identified through several types of analyses, such as protein analysis (*Chen et al., 2012*; *De Canio et al., 2014*) and transcript analysis (*Chen et al., 2014*). As in previous studies (*Chen et al., 2014*; *Chen et al., 2012*; *De Canio et al., 2014*), we used sex-sorted X and Y semen (including ∼90% X(Y) sperm and ∼10%Y(X) sperm ) as the sequencing sample, which may also produce fewer DA miRNAs, due to the inevitable presence of false-negative DA miRNAs in the analytical results. Finally, by employing a conservative approach to identify DA miRNAs (only the DA miRNAs annotated from both Mirdeep2 and Unitas were included in the further analysis), the accuracy of DA miRNA identification was improved, but other DA miRNAs might have been missed.

In the present study, the total numbers of DA and highly abundant non-DA miRNAs were detected from previously reported data obtained from bull sperm (*Capra et al., 2017*; *Sellem et al., 2020*). Moreover, nine highly enriched non-DA miRNAs (bta-miR-100, let-7a-5p,bta-miR-22-3p, bta-miR-151-5p, bta-miR-21-5p, bta-miR-99a-5p,bta-miR-16b, bta-miR-7 and bta-miR-27a-3p) and two DA miRNAs (bta-miR-211, bta-miR-204, and bta-miR-375) were present among the top 20 abundant miRNAs of a previous study (*Capra et al., 2017*; *Sellem et al., 2020*) (Table 2). Some of the differences in sperm miRNA profiles observed in the current study may be due to the use of differences in the treatment of the samples. For example, in *Capra et al. (2017)*'s study, the sperm were cryopreserved in straws for AI, while sperm were stored on dry ice after sorting in the current study. Among the most abundant non-DA miRNAs, bta-miR-100 and bta-miR-151-5p have been shown to be associated with sperm motility in bulls (*Capra et al., 2017*). bta-miR-100, which exhibited the highest abundance in our study, has been previously reported as the most abundant in bull sperm (*Stowe et al., 2014*) and has been identified in sperm of other species: human, pig, and mouse (Table 2) (*Chen et al., 2017*; *Nixon et al., 2015*; *Pantano et al., 2015*). Additionally, bta-miR-100 was shown to exhibit low abundance in the semen of infertile males with semen abnormalities (*Liu et al., 2012a*), suggesting an important role of bta-miR-100 in regulating fertility across mammalian species.

Although sperm miRNA contents have been most extensively explored via high-throughput sequencing in mammals, the function of miRNAs in sperm itself essentially remains controversial. The main cause of this uncertainty is that mature sperm are widely thought to be translationally inactive in the cytoplasm (*Park et al., 2012*; *Rahman et al., 2013*). In a previously reported study of DA sperm proteins and mRNAs, we found that there were no DA proteins corresponding to mRNA which are differentially expressed between X and Y sperm (*Chen et al., 2012*; *De Canio et al., 2014*), which may provide indirect evidence of silencing of translation in sperm. Increasing evidence regarding miRNA-target interactions has revealed a new mode of miRNAs action through which gene translation may be regulated by miRNAs targeting CDS regions (*Hausser et al., 2013*; *Tay et al., 2008*). For instance, inhibition of translation in somatic cells was previously demonstrated to be due

to miRNA binding sites located in CDS regions (*Hausser et al., 2013*). Similarly, miRNAs have been shown to cotarget the 3′ UTRs and CDS regions of maternally expressed mRNAs to regulate embryonic development in early zebrafish embryos (*Hausser et al., 2013*). Furthermore, the application of high-throughput approaches for isolating argonaute-bound target sites indicates that CDS sites are as numerous as those located in 3′ UTRs (*Chi et al., 2009*; *Hafner et al., 2010*). In our study, we found that most sperm mRNAs (82%) and mature oocyte mRNAs (79%) were predicted to be targeted by DE miRNAs through binding to CDS regions (Fig. 1). In addition, the argonaute 2 complex, which is crucial for miRNA function, was found to be bound to miRNAs in mouse sperm (*Liu et al., 2012b*). Here, it is likely that DA sperm miRNAs bind to CDS regions and act as translation-inhibiting factors in sperm (*Hosken & Hodgson, 2014*), and mRNAs are regulated by the CDS regions, which are widespread in sperm and fertilized oocyte. Furthermore, functional analysis of DA miRNA-targeted genes in sperm showed that these genes were involved in nucleosome binding and nucleosomal DNA binding. Mature sperm retains some fraction of residual nucleosomes (*Balhorn, Gledhill & Wyrobek, 1977*). The X chromosome in X sperm was demonstrated to exhibit strong enrichment of nucleosome-binding sites, and the Y chromosome in Y sperm exhibited a strong depletion in bovine sperm (*Samans et al., 2014*). Overall, the results suggest that X and Y sperm, with different sex chromosomes, may contain genes targeted by DA miRNAs that perform different functions in nucleosome binding and nucleosomal DNA binding.

The egg is the ultimate destination for sperm, along with its miRNAs. Mammalian sperm carry subsets of miRNAs into oocyte during fertilization. However, whether sperm miRNAs can play the roles after fertilization is still controversial. One argument relevant to this issue is that the levels of sperm miRNA are low relative to those of unfertilized MII (Metaphase II) oocyte, and fertilization does not alter the MII oocyte miRNA repertoire, suggesting that it plays a limited role in mammalian fertilization or early preimplantation development (*Amanai, Brahmajosyula & Perry, 2006*). However, an increasing number of studies have shown that sperm-borne miRNAs are indeed important for preimplantation embryonic development (*Grandjean et al., 2015*; *Rodgers et al., 2015*; *Sharma et al., 2015*). Shuiqiao Yuan et al. found that sperm with altered miRNAs could fertilize wild-type eggs. However, embryos derived from these partial small noncoding RNA-deficient sperm displayed a significant reduction in developmental potential, which could be rescued by injecting wild-type sperm-derived total or small RNAs into ICSI (Intracytoplasmic sperm injection) embryos, whereas maternal miRNAs were found to be dispensable for both fertilization and preimplantation development (*Yuan et al., 2016*). According to recent studies, even when the content of sperm miRNAs is low, miRNAs can be involved in initiating a cascade of molecular events after fertilization, through targeted degradation of stored maternal mRNAs (*Rodgers et al., 2015*). In addition, the sperm-borne miR-449b can improve the first cleavage division, involve in epigenetic reprogramming and apoptotic status of preimplantation cloned bovine embryos through regulating maternal mRNAs (*Wang et al., 2017*). In the current study, DA miRNAs were more prone to target genes in matured oocyte than in sperm. Taken together, the results seem to indicate that one of the ways in which sperm miRNA perform their roles after fertilization is by regulating maternal

genes. Based on this hypothesis, we carried out the functional analysis of the putative targets of DA miRNAs in the fertilized oocyte. The analysis of GO term annotations indicated that maternal mRNAs in oocyte targeted by DA miRNAs were significantly enriched in 11 catabolic processes. Seven DA miRNAs (bta-miR-127, bta-miR-1271, bta-miR-196a, bta-miR-204, bta-miR-3432a, bta-miR-375 and bta-miR-6529a), along with their 17 target genes were found to be involved in catabolic processes in the mature oocyte (Fig. 3B). Among these miRNAs, miR-204 and miR-375 with their related target genes have been well established to play a clear inhibitory role in catabolic processes in cancer (Lin et al., 2017; Mao et al., 2016). Indeed, spermatozoon fertilization of oocyte was previously demonstrated to trigger a selective process that recognizes and degrades paternally inherited organelles (Al Rawi et al., 2011). Furthermore, X and Y sperm were reported to exhibit different protein profiles (Chen et al., 2012; De Canio et al., 2014) and sex chromosome structures. Based on these enlightening findings, we postulated that, when an X sperm or Y sperm enters the oocyte, sperm-carried DA miRNAs probably have discriminating catabolic functions for X and Y sperm involving different components through regulating related genes. In the present study, thyroid hormone receptor interactor 12 (Trip12), a maternal gene that is putatively targeted by bta-miR-204, bta-miR-1271, bta-miR-375, and bta-miR-3432a, plays an important role in embryogenesis (Kajiro et al., 2011). Moreover, prolonged stress in mice was demonstrated to alter the expression of nine sperm miRNAs, including miR-204 and miR-375, which can regulate maternal mRNAs, resulting in changes in offspring hypothalamic–pituitary–adrenal (HPA) axis responses to stress (Rodgers et al., 2013; Rodgers et al., 2015). Interestingly, one of their targeted maternal mRNAs identified in mouse fertilized oocyte, Serine and arginine rich splicing factor 2 (Srsf2), was predicted to be targeted by another DA miRNAs bta-miR-378 in bull mature oocyte (Rodgers et al., 2013; Rodgers et al., 2015). HPA response patterns differ markedly in males and females (Handa et al., 1994; Kajantie & Phillips, 2006; Kudielka & Kirschbaum, 2005; Verma, Balhara & Gupta, 2011). These finding suggest that bta-miR-204, bta-miR-375 and bta-miR-378, three most significantly DE miRNAs, carried by X and Y sperm to the fertilized oocyte may regulate maternal mRNAs to potentially influence stress reactivity in the offspring. In addition to sperm miRNA, tsRNAs from sperm could act as acquired epigenetic factors and contribute to offspring phenotypes such as metabolic traits . In current study, two tsRNAs (tRNA-Ser-AGA and tRNA-Ser-TGA) were differentially abundant between X and Y sperm. The expression of tRNA-Ser-TGA was shown to be positively correlated with cell proliferation in prostate cancer cell, which could promote the transition of these cells from the gap 2 phase of the cell cycle to the mitotic phase (Lee et al., 2009). This finding suggest the possibility that tRNA-Ser-TGA may act in a similar manner upon delivery to the oocyte (Peng et al., 2012). However, the studies involved in the fields of embryo development related to these two DA tsRNAs remain limited, which still require further exploration.

PiRNAs are small noncoding RNAs that can have significant implications for germ cell development and function. piRNAs are shown to be the most abundant class of small RNAs in human sperm (Pantano et al., 2015). Sellem and colleagues found that 26% of reads were annotated by piRNAs in bull sperm (Sellem et al., 2020). In the present study, the percentage of clean reads mapping to piRNAs database was about 6.8% and 8.7% for

X and Y sperm, respectively (Table S2). The differences in piRNA proportion observed between two studies may be mainly due to the use of different analysis strategies. The high modifications of a single nucleotide at either the 5p or 3p end were the most frequent changes (*Sellem et al., 2020*). In this study, the high threshold that allowed 0 mismatch when mapping the sequences to reference piRNA sequences was used for piRNAs annotation, which would increase the credibility of the conservative piRNA annotated results but decrease the proportion of piRNAs annotated. Genome mapping of such piRNA sequences revealed that piRNAs mostly originate from distinct genome clusters, termed piRNA clusters (*Aravin et al., 2007*), which are a few to hundreds of kb in length. The genomic locations of these loci are often conserved between related species such as mouse and human (*Aravin et al., 2006*; *Gan et al., 2011*), but the sequences of the piRNAs themselves have evolved rapidly differ even between closely related species such as human and chimpanzee (*Lukic & Chen, 2011*). In the present study, one piRNA cluster was reported to conserve deeply in eutherian mammals which are located between the CCAR1 and DDX50 genes were also identified in the present study and were named CXY69, it was conserved in X and Y sperm and also contain STOX1 gene. STOX1 transcript was antisense to the many piRNAs generated in CXY69 cluster (*Chirn et al., 2015*). The different distribution of piRNA clusters, containing different genes and transposons, and abundance of piRNAs between X and Y sperm were identified in this study, suggesting these piRNAs may play different regulatory roles between them. Because of the low level of piRNAs conservation between even closely related species (*Girard et al., 2006*; *Hong et al., 2016*; *Krawetz et al., 2011*; *Lau et al., 2006*), and studies deciphering the functions of piRNAs were still limited, the potential role of they played in X and Y sperm remain to be further understood.

## CONCLUSIONS

In conclusion, the present study revealed the sncRNA contents of X and Y sperm and highlighted the differences in the abundance and diversity of several common sncRNAs across two types of sperm. Additionally, we comprehensively discussed the roles of the DA miRNAs in sperm and fertilized oocyte, which could enhance our understanding of their potential functions involved in sex differences in sperm and early embryonic development.

## ACKNOWLEDGEMENTS

We thank Dr. Denis Larkin and Dr. Marta Farré for valuable comments on the manuscript.

### Funding

This research was supported by a Key Project of Ministry of Science and Technology of China (2017YFD0501903) and the China Scholarship Council (201706760064). The funders had no role in study design, data collection and analysis, decision to publish, or preparation of the manuscript.

## Grant Disclosures

The following grant information was disclosed by the authors:
Key Project of Ministry of Science and Technology of China: 2017YFD0501903.
China Scholarship Council: 201706760064.

## Competing Interests

The authors declare there are no competing interests.

## Author Contributions

- Hao Zhou and Jiajia Liu conceived and designed the experiments, performed the experiments, prepared figures and/or tables, authored or reviewed drafts of the paper, and approved the final draft.
- Wei Sun and Rui Ding performed the experiments, authored or reviewed drafts of the paper, and approved the final draft.
- Xihe Li, Liguo Yang and Shujun Zhang conceived and designed the experiments, authored or reviewed drafts of the paper, and approved the final draft.
- Aishao Shangguan performed the experiments, prepared figures and/or tables, authored or reviewed drafts of the paper, and approved the final draft.
- Yang Zhou, Tesfaye Worku, Xingjie Hao and Faheem Ahmed Khan analyzed the data, authored or reviewed drafts of the paper, and approved the final draft.

## Animal Ethics

The following information was supplied relating to ethical approvals (i.e., approving body and any reference numbers):

The Scientific Ethics Committee of Huazhong Agricultural University approved the experimental design and animal treatments for the present study (permit number: HZAUSW-2017-012), and all experimental protocols were conducted in accordance with the guidelines.

## Data Availability

The sequencing data of X and Y sperm sncRNA are available in the Sequence Read Archive: PRJNA624261.

## Supplemental Information

Supplemental information for this article can be found online at http://dx.doi.org/10.7717/peerj.9822#supplemental-information.

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
