# Peer review of "Differences in small noncoding RNAs profile between bull X and Y sperm"

_PeerJ, doi:10.7717/peerj.9822_

## Round 0.1 · original submission · Major Revisions

Thank you very much for your excellent work. This manuscript has been much improved from the previous version. However, I am writing to let you know that your manuscript has been reviewed by three experts in the field and we request that you make major revisions before it is processed further. Please find your manuscript and the review reports at the following link. I am waiting for your revision version.

Reviewer 1 ·

Basic reporting

DNA data checks
Have you checked the authors data deposition statement?
Yes, the authors indicate the data deposition as accession codes PRJNA624261 (https://www.ncbi.nlm.nih.gov/bioproject/PRJNA624261/) in the GEO datasets in the topic “Preprocessing of small noncoding RNA data” of the revised manuscript.
Can you access the deposited data?
Yes, the above-mentioned data can be easily accessed via https://www.ncbi.nlm.nih.gov/bioproject/PRJNA624261/.
Has the data been deposited correctly?
Yes.
Is the deposition information noted in the manuscript?
Yes, as appears in line 152-155 of the revised manuscript.
1. BASIC REPORTING
1. From the original version of this manuscript, the authors received the comments on professional English language. In the current one, the usage of English language has greatly improved. The proof reading of the manuscript has been made to correct typographical and grammatical errors.
2. The authors indicated the data deposition as accession codes PRJNA624261 in the Sequence Read Archive (SRA). Those raw data are now available through the NCBI Gene Expression Omnibus.
3. Details of the piRNA cluster have been added to Introduction, Methods, and Discussion.

Experimental design

1. The authors have consistently used the term “abundance” in place of “expression” to describe the presence of sperm-borne sncRNA in their data. Similarly, the use of “differential accumulation” instead of “differentially expressed” throughout the revised manuscript is corrected.
2. To verify the enrichment level of X and Y bearing spermatozoa in the current study, the authors have added the statements in line 112-119 of the revised manuscript to show the detailed method of sort reanalysis and thus provided with reference. The use of aforementioned method is certainly recommended, and the study design allows it. The results of resorting procedure are quite acceptable, as shown in Table S1. Using this approach, the replication of information could be achieved by another researcher.
3. The RNA integrity number (RIN) of 2.4-2.6 (in the current study) is quite controversial whether they are already degraded or not. Most RNA-seq experiments rely on the RIN above 5 to be considered as high quality RNAs. From Georgiadis et al (2015), they considered that the mature sperm samples have RIN of 2.7 are poor quality for downstream application. It will be better if the authors could provide some pictures of FastQC for raw sequence data.
Ref: http://dx.doi.org/10.1016/j.juro.2014.07.107

Validity of the findings

1. The real-time PCR experiments (qPCRs) to validate the findings of abundantly presence of miRNAs were conducted and the results are included in the revised version of the paper. The methods (Quantitative real-time PCR (qPCR) validation of the sequencing results) and results are shown in line 210-219 and line 343-348, respectively.

Additional comments

The typo errors of previous version of manuscript have already been solved. However, a few typo and punctuation errors, such as (HOSSAIN et al. 2001; Penfold et al. 1998) of line 50-51; using the sort reanalysis method(Welch & Johnson 1999). of line 111 are found in this version.

Reviewer 2 ·

Basic reporting

Zhou et al described the difference of sncRNA between X and Y sperms of the bull which may play an important role in the regulation of mature mRNA in fertilized oocytes and subsequent embryogenesis. The general content was well organized with professional English. The figure and table here were suitable and sufficient. I find some mistakes of citation in the text, for example, in line 50, (HOSSAIN et al. 2001) should be edited in the proper format. Please carefully check it throughout MS.

Experimental design

The detailed background and methodology gave were sufficient to understand the main objective. I have some suggestions as below;
(i) The information of bull used in this study should be given in necessary detailed such as age or breed because these factors may influence the DA.

(ii) The author employed NGS with the BGISEQ-500 platform for sequencing; however, the author should provide the reference of the platforms used. Also, RNA integrity must be fall in the high-quality criterion for doing NGS. The author should provide the information on RNA integrity obtained from this investigation in MS. Please specify where these data were detailed in Table S1?

(iii) the author did validation of sequencing result by qPCR for four miRNAs including bta-miR-204, bta-miR-3432a, bta-miR-652, and bta-miR-378. The author gave the information of primers for these miRNAs but I note the primers of each miRNA have only an oligonucleotide. Should they have forward and reverse sense? please verify them.

Validity of the findings

The finding of this study was well-written with a logic style. I have some suggestions a bit in the results section. I find the author provides few discussion points in the result context, for example, in line 270-271, “The RNA integrity number (RIN) was approximately 2.5, which was consistent with previous studies (Mao et al. 2014; Sendler et al. 2013).” and in line 286-288 “A considerable portion of the sperm small noncoding RNA could not be annotated in existing public databases, which was consistent with a previous study (Krawetz et al. 2011).”. These statements should be moved in the discussion parts.

Besides, the author found the difference in tsRNA between X and Y sperm, resulting in tRNA-Ser-AGA and tRNA-Ser-TGA. In the discussion, why this point has not been mentioned?

Additional comments

(i) The scientific name must be written in an italic style, please check in the supplemental table.
(ii) The abbreviation should be written with full name in the first place such as ICSI and MII in the discussion.

Reviewer 3 ·

Basic reporting

The role of sperm borne sncRNAs are more and more studied as key elements for embryo development, in many species. In bulls, more results have to be produced to paint a clear overview of the content/variation/involvement of these sncRNAs in different biological pathways.
Establishing a comparison of sncRNAs content of X and Y sperm cells, represent an interesting way to increase our knowledge around these elements. Some minor modifications have to be brought to your manuscripts, but to my point of view also some important lacks have to be filled in your study (as the used of adjusted or corrected p values instead p values). The number of significant miRNAs will change and potentially the interpretation of results.

Experimental design

Line 103 : The number/origin of sample are not clear. You talked about 3 bulls, but how many samples? 6 (X and Y fractions for each bulls) or 2 (X and Y fractions polled) ?

Line 124 : “In brief, the samples were thawed on ice and added to TRIzol…..”
Do you started from frozen semen? It was not mentioned on the previous paragraph (semen collection).

How many sperm cells have been used for the total RNA extraction?

According to several works published recently in bovine (example: Sellem et al., 2020), the lysis step of bull sperm cell is quite difficult, and Trizol alone is not enough to lyze properly the cell. Have you met some difficulties with your approach? Have you some pictures of the sperm cell after the treatment?

Line 162 : “…putative known mature miRNAs…”
You have to add the word “and” between putative and known

Line 183: “…By applying thresholds of a P-value < 0.05….”
: you have qualified miRNAs as “differentially expressed” based on the FC and the p-value. Due to the statistic hazards on important number of tests, you will be able to detect a false significant tests (p-value < 5%)…that why it’s important to use the adjusted p-values and not just the p-values. Please to re-analyse your data under this new approach.

Line 187: “Functional analysis of DA miRNAs”.
May I ask you to change the word “analysis” by “annotation”? The work done here was only in silico, while the expression “functional analysis” tent to thing about real lab work.

Validity of the findings

Line 227: “…indicating a lack of foreign RNA in sperm RNA samples…”
Indicating a lack of intact foreign RNA….

Lines 343 to 348: May I ask you to put this paragraph before the one about the identification of DA miRNA?

May I ask you to be less “positive” about the concordance of results between the two technologies? The results for the miR-204 and 652 are not so equal…and they represent the half of your tests.

Line 242: “…A considerable portion of the sperm small noncoding RNA could not be annotated in existing public databases, which was consistent with a previous study (Krawetz et al. 2011)…”
May I ask you more details? “A considerable portion” means how many in your case?
The Krawetz et al., work was published in 2011…The reference database (miRbase) has been updated. May I ask you to compare your results with another publication (example: Sellem et al., 2020)?

Lines 245 to 247: “…In total, 490 and 202 known Bos taurus miRNAs were detected by Unitas and Mirdeep2, respectively… »
It’s not clear….May I ask you to rephrase these 2 sentences?
Lines 247 to 259: Once again, the use of “p value” is not recommended due to the high number of tests…You need to use the “adjusted p value” (whatever the correction applied).

Lines 297 to 313: The number of targets could be very important per miRNA. Do you have applied of filter to choose the most relevant ones?


Discussion
Lines 353 to 357 : “…Previous studies have revealed that adjacent sperm cells can share gene products through intercellular bridges during spermatogenesis, suggesting that miRNA molecules may be shared between X and Y sperm cell during spermatogenesis (Fawcett et al. 1959), which may explain why a part (118) of non-DE small RNAs were identified between X and Y sperm…”
This supposition is too hazardous. Several other explanations could be involved...May I ask you to change your sentences or to bring more details?

Lines 368 to 383: New studies have been published between 2017 and 2020, and especially Sellem et al.,2020 in bull semen. May I ask you to compare also your results to theirs?

Line 413 to 450: We would expect more discussion about the differences between X and Y sperm cells. You have highlighted several interesting genes with your KEEG analysis. So please may I ask you to bring more information about their potential role in embryo development (in a X and Y context)?

Lines 451 to 467: The number of piRNAs identified in your studies is quite different from those obtained in another ones. May I ask you to add some comparisons with freshly published studies?

What about the tsRNAs? May I ask you to add a paragraph in the discussion part?

Conclusions
You mainly discussed about miRNAs…It’s just a suggestion but maybe you should change the term “sncRNA” into “miRNAs”.

---

## Round 0.2 · accepted · Accept

Thank you very much for your scientific contribution. I hope you will consider publishing good research with us in the future.

Reviewer 1 ·

Basic reporting

1. The typographical and grammatical errors are carefully rechecked by the authors. No obvious errors are found in MS_R2.

Experimental design

1. From last comments delivered to authors concerning the RNA quality, the plots of per base sequence quality of raw sequencing data by FastQC are given as Additional material 1 in the current revision. Entire raw sequence reads (n = 6) have Phred quality scores above 30 represent the estimated probability of very small error in base calling. Then, the results in this study are reliable. Detailed information on FastQC could be found in the topic “Preprocessing of small noncoding RNA data” in M&M and “Read counts of each RNA class” in the results. I believe that most of my concerns expressed in the previous review has been handled.

Validity of the findings

I have no additional comments.

Reviewer 2 ·

Basic reporting

no comment

Experimental design

no comment

Validity of the findings

no comment

Additional comments

The manuscript meets the criteria to accept for publishing in PeerJ.

Reviewer 3 ·

Basic reporting

no comment

Experimental design

no comment

Validity of the findings

no comment

Additional comments

no comment